# A Review of Voice-Based Pain Detection in Adults Using Artificial Intelligence

**DOI:** 10.3390/bioengineering10040500

**Published:** 2023-04-21

**Authors:** Sahar Borna, Clifton R. Haider, Karla C. Maita, Ricardo A. Torres, Francisco R. Avila, John P. Garcia, Gioacchino D. De Sario Velasquez, Christopher J. McLeod, Charles J. Bruce, Rickey E. Carter, Antonio J. Forte

**Affiliations:** 1Division of Plastic Surgery, Mayo Clinic, Jacksonville, FL 32224, USA; 2Department of Physiology and Biomedical Engineering, Mayo Clinic, Rochester, MN 55902, USA; 3Department of Cardiovascular Medicine, Mayo Clinic, Jacksonville, FL 32224, USA; 4Department of Health Sciences Research, Mayo Clinic, Jacksonville, FL 32224, USA

**Keywords:** machine learning, pain assessment, voice analysis, vocal biomarkers

## Abstract

Pain is a complex and subjective experience, and traditional methods of pain assessment can be limited by factors such as self-report bias and observer variability. Voice is frequently used to evaluate pain, occasionally in conjunction with other behaviors such as facial gestures. Compared to facial emotions, there is less available evidence linking pain with voice. This literature review synthesizes the current state of research on the use of voice recognition and voice analysis for pain detection in adults, with a specific focus on the role of artificial intelligence (AI) and machine learning (ML) techniques. We describe the previous works on pain recognition using voice and highlight the different approaches to voice as a tool for pain detection, such as a human effect or biosignal. Overall, studies have shown that AI-based voice analysis can be an effective tool for pain detection in adult patients with various types of pain, including chronic and acute pain. We highlight the high accuracy of the ML-based approaches used in studies and their limitations in terms of generalizability due to factors such as the nature of the pain and patient population characteristics. However, there are still potential challenges, such as the need for large datasets and the risk of bias in training models, which warrant further research.

## 1. Introduction

### 1.1. Background

It has been established that pain represents a significant issue in the realm of public health within the United States. However, its subjective nature is a challenge for effective pain management [1,2,3]. The International Society for the Study of Pain (IASP) defines pain as “an unpleasant sensory and emotional experience associated with, or resembling that associated with, actual or potential tissue damage” [4,5,6]. The previous definition was revised in light of the many variables that affect how intense one’s painful episodes are, including past painful experiences, cultural and social contexts, individual pain tolerance, gender, age, and mental or emotional state [2,7].

Despite a thorough understanding of the pathophysiological processes underlying the physical pain response and the technological advances to date, pain is often poorly managed. Misdiagnosis of pain levels through associated subjective biases, as it is currently performed, can increase unnecessary costs and risks. In addition, poor pain relief can result in emotional distress and is linked to several consequences, including chronic pain [8,9,10].

Self-reports and observations form the basis of well-defined instruments for measuring pain and pain-related factors. Visual analog scales (VAS), the McGill Pain Questionnaire, and the numeric rating scales (NRS) are examples of patient-reported outcome measures (PROs, self-reports, or PROMs), which are frequently considered “the gold standard” for measuring acute and chronic pain. As these strategies depend on patients’ accounts, they are exclusively applicable to individuals with no verbal or cognitive impairments [1,6,11,12]. Because of this, all of these established techniques are inapplicable to newborns, delirious, sedated, and ventilated patients, or individuals with dementia or developmental and intellectual disorders [13]. Such patients are entirely dependent on others’ awareness of nonverbal pain cues. In light of this, observational pain scales are advised for use among adults in critical conditions when self-reported pain measurements are not feasible. Even so, the reliability and validity of these instruments are still limited, because even qualified raters cannot ensure an unbiased judgment [5,14,15,16,17,18,19].

Automatic pain recognition has transitioned from being a theory to becoming a highly important area of study in the last decade [10]. As a result, a few studies have concentrated on utilizing artificial intelligence (AI) to identify or classify pain levels using audio inputs. While face and body occlusions are frequently observed in newborns, these methods are especially relevant for examining their cries as the more accurate way to identify neonates’ discomfort [20]. However, there remains a significant disparity in voice-based pain detection among adult patients.

### 1.2. Research Question and Objectives

This study aimed to provide an updated and concise view of the use of AI with voice or audio to detect pain in adult patients. Specifically, this study aimed to (1) summarize the current state of new research in this area, (2) identify the limitations and challenges of existing approaches, and (3) propose potential solutions and future research directions to address these limitations and challenges.

### 1.3. Methodology

The literature search was performed in the PubMed, Medline, Scopus, Web of Science, CINAHL, and Google Scholar databases. We included all the studies on the use of AI with voice for pain detection in adults up to 2023. Studies that were published in the English language and peer-reviewed journals were included.

#### 1.3.1. Inclusion and Exclusion Criteria

We included studies that focused on the use of AI algorithms to analyze vocal features of pain in adult human subjects. The studies used a variety of AI techniques, including deep learning, natural language processing, and machine learning. They also made use of various vocal samples, such as speech, moans, and groans. Studies were excluded if they only focused on animals, infants, pediatrics, and non-verbal or unconscious patients.

#### 1.3.2. Search Strategy

The search was conducted using a combination of keywords and controlled vocabulary. The search terms were related to AI, pain, voice, and detection. The search was restricted to human research and articles in English. Two reviewers independently screened the search results, while a third reviewer resolved any conflicts.

#### 1.3.3. Data Extraction and Analysis

Two independent reviewers extracted the data using a standard data extraction form. The extracted information covered the study’s methodology, sample size, type of ML employed, categories of speech and pain qualities examined, speech parameterization method, model validation method, and AI algorithm accuracy.

We begin by providing a general review of the use of voice to identify pain (Section 2) and the role of artificial intelligence in pain detection (Section 3). In Section 4, we review the relevant papers, and in Section 5, we discuss the limitations and difficulties of AI models for pain detection using voice. In Section 6, the paper is concluded.

## 2. Voice, Pain, and Artificial Intelligence

### Using Voice to Detect Pain

To extract speech-specific features from the recorded voice, analysis techniques such as perceptual linear prediction (PLP), PLP-relative spectra (PLP-RASTA), and mel-frequency cepstral coefficients (MFCC), are required.

These techniques can be used for various tasks such as speech recognition, speaker identification, and emotion recognition. An unknown sample can be predicted by training a neural network with feature vectors, while classification and recognition can be performed using techniques such as Support Vector Machine (SVM), the Hidden Markov Model (HMM), Dynamic Time Warping (DTW), and Vector Quantization (VQ) [21,22,23,24,25].

According to the studies, verbal or acoustic cues can be used to identify and even categorize pain according to its intensity level. The mean and range of voice fundamental frequency (F0), loudness, and degree of periodicity are some of the acoustic characteristics of voice that have been shown to indicate pain levels. Pain-related vocalizations, such as screaming, groaning, moaning, and crying, are often accompanied by changes in these phonetic features and can be suggestive of pain [26,27,28,29,30].

Several studies on the use of voice for pain detection were reviewed by Helmer et al. in their 2014 paper [15]. Their research focused on investigating whether vocalizations alone could indicate the presence of pain.

Of the 35 studies that were examined, 29 of them provided evidence of a connection between vocalization and pain.

Increased pitch and louder vocalization were shown to be more closely related to pain than other phonetic traits.

They showed that in children who can only cry to express themselves, crying might be more related to distress than showing pain. However, overall, children’s crying was proven to be a pain indicator.

Most studies found a link between moaning and pain, whereas screaming or verbalizing was only sometimes associated with pain. Additionally, groaning and sighing were frequently not correlated with pain.

The review also showed that the individual’s personality and social context could impact communicative pain behavior. Therefore, vocalization cannot be solely attributed to pain. Still, considering the significance of distress and social interaction, it is reasonable to infer a correlation between vocalization and pain.

## 3. Artificial Intelligence in Pain Detection

### 3.1. AI Techniques Used in Pain Detection

Although breakthroughs in computing have previously been observed in regard to medical applications, it can be challenging to distinguish between various types of pain [14]. Chronic pain is more difficult to diagnose and necessitates the recognition of more subtle pain coping mechanisms, but the accompanying facial, behavioral, and verbal pain indications make it quite simple to recognize acute pain [31].

Artificial intelligence (AI) describes a system’s ability to mimic human behavior and exhibit intelligence, which is now viewed as a branch of engineering. The adoption of low-cost, high-performance graphics processing units (GPUs) and tensor processing units (TPUs), quicker cloud computing platforms with a high digital data storage capacity, and model training with cost-effective infrastructure for AI applications have all contributed to AI’s unprecedented processing power [32,33,34,35].

There are two main categories of AI applications in the field of medicine: physical and virtual. The virtual subfield involves machine learning (ML), natural language processing (NLP), and deep learning (DL). On the other hand, the physical subfield of AI in medicine involves medical equipment and care bots (intelligent robots), which assist in delivering medical care [36,37,38,39].

Because of their cutting-edge performance in tasks such as image classification, DL algorithms have become increasingly prominent in the past ten years. The ability of a machine to comprehend text and voice is described as NLP. NLP has many practical applications, including speech recognition and sentiment analysis.

Machine learning algorithms may be classified into three categories: unsupervised (capability to recognize patterns), supervised (classification and prediction algorithms based on previous data), and reinforcement learning (the use of reward and punishment patterns to build a practical plan for a specific problem space). ML has been applied to a variety of medical fields, including pain management [37,39,40].

To establish an automated pain evaluation system, it is essential to document the pertinent input data channels. Modality is the term used to describe the behavioral or physiological sources of information. The principal behavioral modalities are auditory, body language, tactile, and facial expressions [20,41].

The availability of a few databases with precise and representative data linked to pain has allowed for recent developments in the field of automatic pain assessment [42], with most works focusing on the modeling of facial expressions [7].

Numerous databases have been established by pooling data from diverse modalities acquired from distinct cohorts of healthy individuals and patients. Most studies use the following publicly available databases: UNBC-McMaster, BioVid Heat Pain, MIntPAIN, iCOPE, iCOPEvid, NPAD-I, APN-db, EmoPain, Emobase 2010, SenseEmotion, and X-ITE.

The BioVid Heat Pain Database is the second most frequently used dataset for pain detection after the UNBC-McMaster Shoulder Pain Archive Database. In the first case, video recordings of 90 healthy people under the influence of heat pain applied to the forearm are collected. The second is made up of 200 videos of participants’ faces as they experience pain through physical manipulation of the shoulder [20,43,44,45].

Voice, meanwhile, has thus far received little consideration. Out of all the publicly available databases, BioVid Heat Pain, SenseEmotion, X-ITE, and Emobase use audio as one of their modalities. To fully leverage the potential of these databases, it is imperative to employ deep learning algorithms. Systems built on deep learning operate in two stages: training and inference. During training, the system is presented with a large dataset so as to teach it to recognize patterns and make predictions. Then, the trained model is used for inference, creating predictions based on the new data [20,43,44,46].

### 3.2. AI Models Used in Pain Detection from Voice

Multiple artificial neurons are merged to create an artificial neural network (ANN). These artificial neurons imitate the behavior and structure of biological neurons. Furthermore, to enhance the efficiency and precision of the ANN, the neurons are organized into layers for ease of manipulation and precise mathematical representation.

The operation of artificial neurons is governed by three fundamental principles: multiplication, summation, and activation. Initially, each input value is multiplied by a distinctive weight. Subsequently, a summation function aggregates all the weighted inputs. Finally, at the output layer, a transfer function transmits the sum of the previous inputs and bias [47,48].

The topology of an ANN refers to the way in which different artificial neurons are coupled with one another [47]. Different ANN topographies are appropriate for addressing different issues.

The most critical topologies are as follows:(1)Recurrent Artificial Neural Networks (RNNs).(2)Feed-Forward Artificial Neural Networks (FNNs).(3)Convolutional Neural Networks (CNNs): These use multiple layers to automatically learn features from the input data.(4)Long Short-Term Memory (LSTMs): It can handle vanishing and exploding gradients, which are common problems in the training of RNNs.(5)Multitask Neural Network (MT-NN): This employs the sharing of representations across associated tasks to yield a more advanced generalization model.

Bi-directional Artificial Neural Networks (Bi-ANN) and Self-Organizing Maps (SOM) can be named as other types [47,48,49].

An artificial neural network can solve a problem once its topology has been selected and it has been tuned and learned the right behavior [47,48].

Lately, there has been significant focus on artificial intelligence algorithms in sound event classification and voice recognition, as shown in Figure 1.

Keyword spotting (KWS), wake-up word (WUW), and speech command recognition (SCR) are three essential techniques in speech processing that enable machines to recognize spoken words and respond accordingly.

Keyword identification technology is an automated approach to recognizing specific keywords within an uninterrupted spoken language and vocalization flow.

KWS systems are less reliant on high-quality audio inputs. They are created to be cheap and flexible and to run accurately and reliably on low-resource gadgets such as embedded edge devices [50,51,52,53].

Researchers have begun to create algorithms in order to automate pain level assessment using speech due to developments in signal processing and machine learning methods. As an example, Tsai et al. [14] employed bottleneck LSTM to detect pain based on a subset of the triage dataset, specifically prosodic signals. Later, Li et al. [7] introduced age and gender factors into a variational acoustic model.

## 4. Review of the Studies

### 4.1. AI and Pain Triage

Acute and chronic pain management differ significantly in the respective approaches. While acute pain management can be addressed using various medication options ranging from over-the-counter drugs to opioids, chronic pain management primarily relies on psychiatric interventions to control the underlying cause, which may sometimes even require surgical interventions. The authors of [54,55,56] conducted a study on chronic pain detection and followed patients with spinal/brain injuries for six months. In contrast, the authors of [50] focused on acute MI pain and achieved the highest overall accuracy using their CNN model. Other studies [14,57] followed patients for one hour to assess their pain levels and conduct an additional audio session, suggesting that the nature of their pain was acute (Table 1).

Accurate pain assessment is critical for effective pain management, and improvements in the precision of pain assessment can reduce the risk of incorrect management, especially in opioid prescription, which carries a risk of addiction and can pose a threat to patients’ lives [8,58,59].

In 2016, Tsai et al. [57] conducted a study evaluating their framework against the NRS. They used the audio–video system in combination with the NRS and various measures (age, vital signs, pain levels, pain levels predicted using SVM models), as well as the two clinical outcome variables of painkiller prescription and disposition. The preprocessing steps undertaken in the study comprised manual segmentation of the recorded data followed by the extraction of LLDs. Subsequently, z-normalization was applied to each patient. Given that the duration of each session was 30 s, two encoding methods were employed to create a constant-length feature vector for each session: the computation of statistical functionals on the LLDs and adoption k-means bag of word (BoW) coding.

Linear regression models were trained using these measures to generate scores for each outcome. The study found that even when considering the best available medical instruments, such as physiological data and the NRS, the use of audio–video-based pain level assessment improves the accuracy in predicting whether a doctor will prescribe analgesics or hospitalize the patient.

The method of “embedding stacked bottleneck features”, a specific type of deep neural network architecture, provides a major advantage in audio processing. They are computationally more efficient and significantly enhance the performance of AI models when confronted with complex data containing numerous variables [14,60,61,62].

Tsai et al. conducted one of the first and most influential studies in the field of AI use to detect pain with voice in real adult patients in 2017 [14]. They employed a stacked bottleneck LSTM model, intending to classify the patients’ pain levels.

Patients who initially presented with symptoms of chest, abdomen, limbic pain, and headaches had their NRS pain scores recorded for the follow-up period. During these sessions, nurses conversed with the patients regarding their pain location and characteristics.

Prosodic (pitch, intensity, etc.) and spectral (frequency content of the audio signal) elements were extracted from the vocal data to create two different subgroups. The authors created a compressed data representation with reduced dimensions and utilized stacked bottleneck acoustic attributes with the LSTMs to generate deep bottleneck features (DBFs).

The DaAi database was used to train an unsupervised LSTM sequence-to-sequence autoencoder. After that, the authors fine-tuned the stacked bottleneck layer by encoding the patients’ sentences through different network layers. Lastly, they used the SVM to classify the pain levels. They showed that this novel approach could obtain a good accuracy in binary (severe, mild) and tertiary (severe, moderate, and mild) pain classification tasks.

The regular monitoring of individuals with heart conditions and provision of immediate medical attention can prevent emergencies from occurring.

For example, over 80% of individuals who experience a heart attack report a tight, squeezing, or crushing sensation in the chest, which can also radiate to the arms, shoulders, neck, jaw, or back [50,63].

The main objective of the study of Mohan et al. in 2021 was to design an audio-based intelligent system to recognize distress signals during MI in an interior environment with a low-power embedded GPU (graphics processing unit) edge device coupled with deep learning (Jetson Nano).

This device is small and portable. It includes a GPU for conducting accelerated computing operations at a network’s edge rather than in a centralized data center or cloud environment.

The authors instructed 60 people to scream and use the Kannada language to communicate as if they were having a heart attack.

The study included three stages: In stage 1, audio data samples were gathered and prepared. A CNN model was trained during stage 2, and stage 3 involved the deployment of the model trained in Jetson Nano. Finally, the researchers also tested the CNN model on a separate, private audio dataset to validate their approach.

Overall, the study results suggest that the proposed supervised audio classification scheme can effectively detect sound distress in an indoor smart space, even in challenging environments with background noise [50].

### 4.2. Pain as a Complex Affection

The experience of pain is intrinsically linked to an individual’s emotional state, which can manifest in various forms of expressive behavior. Research has indicated that emotions are instrumental in shaping both the pain experience and the associated behavioral responses [64,65,66,67]. However, there has not been as much focus on understanding how a person’s internal emotional state influences the pain they report.

In 2019, two studies were carried out by Tsai et al. to examine the link between self-reported pain levels and emotional states (Experiment 1) as well as the efficiency of an emotion-enriched multitask network (EEMN) in enhancing pain level detection by incorporating emotion-related data (Experiment 2). They also used ‘activation’ as the level of energy and intensity of the emotion and ‘valence’ as the negativity or positivity of the emotional expression.

Patients with headaches and abdomen, chest limbic, and lower back discomfort from the emergency department’s triage were included. The information included vital signs, audio–visual recordings, and other relevant outcomes.

The findings indicated a substantial inverse relationship between valence mood and pain severity, while no such relationship existed between activation and the pain level [2].

For each recorded sample, the authors obtained acoustic characteristics utilizing the eGeMAPS feature set available in the openSMILE toolkit.

In the second trial, a multimodal fusion model’s capacity for pain level recognition was enhanced using the EEMN. The model was trained with a multitasking network to focus on the primary objective of pain intensity and an auxiliary task of either valence or activation. By adding valence as an additional task to detect emotions, the study found that both speech and facial modalities had generally high recognition rates for pain levels.

### 4.3. Can the Speech Prosody Act as a Biosignal?

Various studies have established a correlation between pain and physiological indicators, such as systolic blood pressure and heart rate. Moreover, research has highlighted the influences of physiological factors on human vocalization [68,69,70,71,72,73,74].

In 2016, Oshrat et al. [54] conducted a study to determine whether the pain level can be measured based on voice biosignal parameters. The authors interviewed 27 adult participants with spinal cord or brain injuries and injury-related pain. Data were collected over six months through short voice-recorded interviews, and pain levels were evaluated using NRS. The voice samples were processed through manual cutting and feature extraction utilizing openSMILE software.

Machine learning analysis was performed using the WEKA suite, and the classifier SMO was used for classification. In addition, the authors developed two features: sequence indication and heat map thresholding.

Sequence indications refer to methods of analyzing sequences of values to identify patterns or trends in data, which can be used to measure changes in voice parameters as an indication of pain.

The authors created two different lists of sequence-length value indicators for audio samples. Then, they deployed these indicators in the PLP analysis of speech to investigate variations in the voice characteristics’ pace of change. Next, they developed a feature extraction technique called heat map thresholding, which involved the display of coefficients extracted from a voice sample in a two-dimensional matrix and the production of heat map images. They compared pairs of such images of participants with low and high reported pain levels and observed differences in the color patterns. Finally, they ran two classification experiments of “no significant pain” versus “significant pain”. In the first case, only the OpenSMILE features were retrieved; in the second, they added newly generated features.

The study indicated a correlation between the concurrent self-reported pain intensity and the quantifiable speech biosignal attributes, similar to the findings of the other two studies which demonstrated the effectiveness of prosodic low-level descriptors for spectral LLDs in identifying pain [14,57].

### 4.4. Age, Gender, and Pain

Recent research has indicated that ethnicity, race, gender, and age may all impact how someone experiences pain. For example, women tend to encounter more intense pain that occurs more frequently and lasts longer than men. While evidence suggests a decline in pain perception as individuals age, older adults may experience heightened sensitivity to pressure-induced pain [75,76,77,78,79,80].

Li et al. [7] proposed a method for automatic pain level recognition that involves the learning of latent acoustic representations embedded with attributes of gender and age. They achieved this by utilizing a conditional variational autoencoder optimized with the criterion of the maximum mean discrepancy (MMD-CVAE). For the preprocessing task, the authors utilized the acoustic LLD set called eGeMAPS. The features were subjected to z-normalization for every speaker, known as speaker normalization. Subsequently, context window expansion was implemented. The acoustic LLDs were also transformed into hidden features using a CVAE.

Typically, personal attribute dependency in pain recognition is addressed by training multiple independent models, such as gender-specific or age-specific models. However, the authors’ proposed MMD-CVAE approach directly embeds this information in the encoded acoustic space.

The authors demonstrated that, compared to females, male patients with severe pain express their pain with limited maximum jitter (differences in the pitch period of a speech waveform). However, female voices are more variable in pitch and have less harmonic content than male patients. In addition, elderly patients’ voices, in the context of pain, are less variable in pitch and amplitude than non-elderly voices.

The authors also showed that encoding the LLDs for latent representation using VAEs can improve the recognition UAR compared to other models that rely only on functionals.

They also demonstrated the effectiveness of their proposed approach using a large pain database of real patients and achieved significant improvements in the classification accuracy compared to existing methods. Their findings highlight the potential of using MMD-CVAE for personalized pain recognition and the importance of considering personal attributes in pain perception research.

## 5. Limitations and Challenges of AI Models in Pain Detection from Voice

The difficulty of automatic pain recognition arises from its intricate and subjective nature, which is susceptible to multifaceted influences, including but not limited to an individual’s personality, social environment, and past experiences. This matter remains the subject of ongoing scholarly discourse [81].

In comparison to healthy adults, people who are younger, older, in the intensive care unit, or have dementia have different pain thresholds and levels of pain tolerance. Previous studies have found that physiological indications can distinguish between “no pain” and “pain”. Still, researchers have had difficulty in differentiating between pain levels [43,54,69,70,71,75,77,78,80,82]. Moreover, nonverbal expressions of pain may not necessarily be consistent with verbal reports of pain [83].

Deep learning algorithms have a significant advantage in representing complex problems and subjective measurements, such as pain.

However, the majority of the research in this field has been on exploiting visual clues, including facial expressions, to identify pain.

Even though vocalizations are a well-recognized sign of pain and are frequently utilized in clinical contexts to measure pain intensity, the application of AI with voice or audio to identify pain has received much less attention. The difficulties in interpreting voice or audio data may be one factor affecting the limited number of studies in this area. For instance, the pitch, length, and other acoustic characteristics of pain vocalizations can vary greatly, making it challenging to create robust algorithms for pain detection based on these signals. Additionally, it can be challenging to separate pain vocalizations from other voices due to background noise and other confusing elements. For example, the authors of [50] showed that the limitation in automatic MI detection was the fact that their obtained model needs to distinguish between speech and various pain states in order to minimize false alarms.

In all of the studies regarding the detection of pain using AI, the accuracy of pain measurement could be improved by creating a combined multimodal framework that incorporates facial expressions, body language, biological signals (ECG, EMG, and GSR of the trapezius muscle), audio, and vocalization data. Furthermore, our detailed comprehension of the described medical emergency application could be improved by using a large number of validated datasets [81]. This is similar to the study in [50], in which the broadening of the database could lead to improved classification outcomes that could be applied to real-world situations.

Pain is influenced by different personal and social circumstances, which is another reason explaining why there is not more research in this field. This variability can make the development of accurate and generalizable algorithms for the detection of pain challenging.

Though the 13 findings of Tsai et al.’s study were encouraging, there were still constraints on the scope of the existing databases, specifically in regard to the collection of the actual triage sessions and the use of the DaAi background corpus to enhance the behavior patterns noted and represented in their framework [14].

In [50], the researchers had difficulties in creating and developing a dataset for automatic pain identification that fulfills benchmarking norms. This highlights the urgent need for a comprehensive reference dataset of various ethnicities and languages to reliably determine pain levels using audio signals. The inclusion of diverse voices with background noises and sounds is also essential to ensure that the model is reliable and can be generalized to various scenarios.

## 6. Conclusions

### 6.1. Summary of Findings

In 2016, Tsai et al. [57] found that combining the audio–video-based pain scale with other clinical measures results in more accurate analgesic prescription and patient disposition predictions than using the NRS scale alone. Additionally, in 2019, in emergency triage, where there is natural spoken contact between a medical practitioner and a patient, the authors of [14] concentrated on using machine learning algorithms to estimate the pain intensity. A similar process was carried out in [2], whose authors created an emotion-enriched multitask network (EEMN) by simultaneous optimization using effect states and multimodal behavior data to enhance self-reported NRS classification (acoustic and facial expressions). They showed that pain can be assumed to be a complex affect, and the use of AI to detect pain with this new definition can change traditional pain assessments. Additionally, the CNN method was employed for audio-based emergency detection during MI by the authors of [50]. They demonstrated that their lightweight CNN architecture was well-adapted to deliver an immediate response to heart vital attack signs in real-time processing. Oshrat et al. [54] discovered that in addition to the patient’s report, speech in pain could be obtained as a biosignal for categorizing pain episodes. They successfully classified the pain levels into “no severe pain” and “severe pain” using auditory characteristics and heat map threshold image analysis for the first time.

The study conducted by Li et al. in 2018 proposed a novel approach for recognizing pain levels using vocal-based information. The approach involved embedding personal attributes such as age and gender directly into the latent layer of a machine learning network called MMD-CVAE. This enabled the encoding of acoustic LLDs into the latent representation, improving pain level recognition accuracy [7].

Mohan et al. [50] achieved the highest accuracy of 97.91% using the CNN algorithm, while Tsai et al. [57] obtained the lowest accuracy for tertiary pain classification using the SVM and linear regression models (51.6%). Oshrat et al.’s CSF and SMO model showed a promising increase in the CCI ratio after the addition of new features to openSMILE [54]. The authors of [2,7] achieved a UAR of over 70% for binary pain classification using MMD-CVAE and EEMN, respectively. However, they showed weaker results in tertiary classification due to its more ambiguous nature, with an accuracy ranging from 47.4% to 52.1%.

### 6.2. Clinical Applications

The results of this review have essential and wide-ranging potential implications. According to the studies we evaluated, it is possible to efficiently and accurately identify various forms of pain by combining vocalizations and audio recordings with cutting-edge machine learning algorithms.

These discoveries may lead to new instruments and technologies that use voice analysis to identify and manage pain. In healthcare settings, where precise and prompt pain identification is essential for optimal patient treatment, such technologies could be very helpful. In addition, voice-based pain-monitoring devices could be integrated into surveillance systems for patients who cannot self-report pain, such as nonverbal or comatose patients.

Another implication is the employment of voice-based pain detection systems in research settings to better comprehend the complicated nature of pain.

Overall, the results of the research that has been reviewed have significant implications for the creation of new technologies and therapies that make better use of voice and machine learning to detect and control pain. While AI-based tools can reduce healthcare costs and save time, these effects may significantly influence pain treatment and the lives of patients with acute and chronic pain.

### 6.3. Potential Areas for Future Research

Upcoming research utilizing voice recognition and AI to identify pain offers various exciting directions for investigation.

First, to construct AI models, it is crucial to consider pain as an emotive experience. Since pain is a subjective experience, future research may benefit from a better comprehension of the subtleties of pain perception and the significance of accurately capturing the complexity of this experience.

Secondly, the precision and efficiency of prediction models could be greatly improved by combining several modalities and thorough data collection, including personal attributes.

Moreover, expanded objective metrics for the assessment of AI models are required. For instance, to improve the accuracy of pain identification by voice analysis, future studies might investigate the combination of additional metrics, such as voice, with physiological parameters such as the skin conductance level or heart rate. Additionally, the inclusion of routine practice scenarios could offer a more realistic environment for evaluating the efficacy of AI models in pain detection.

Finally, there is still a shortage of studies on the use of AI for pain recognition in adult populations, despite the rising corpus of literature on the topic in pediatric and neonatal populations. It is therefore essential to address this knowledge gap and broad the research focus to include adult populations. Future research could then contribute to the creation of focused strategies to enhance pain management in clinical settings and offer a more thorough understanding of the effectiveness of AI-assisted pain detection in various patient populations.

### 6.4. Limitations

The main drawback of our review is the lack of information about the use of speech and audio modalities in adult patients in real-world settings. Since engineers and scientists have only recently become interested in this field of inquiry, there are still plenty of prospects for additional study and improvement. However, the small amount of research that has been performed on this subject makes it difficult to comprehend all its subtleties and complexities in full. In addition, although ethical considerations represent a crucial area of focus in AI research, this review did not cover them due to its broad scope.

### 6.5. Final Remarks

Despite the challenges, there is growing interest in the utilization of speech or audio with AI to detect pain, as it could have a wide range of clinical and research applications. For example, it can be used to track patients with chronic pain or to evaluate pain in patients who cannot express it verbally. However, further research is needed in order to develop and assess precise algorithms that can recognize pain in adults through voice or audio signals. Additionally, more investigation is necessary in order to understand the factors that affect the correlation between vocalizations and pain.

## Figures and Tables

**Figure 1 bioengineering-10-00500-f001:**
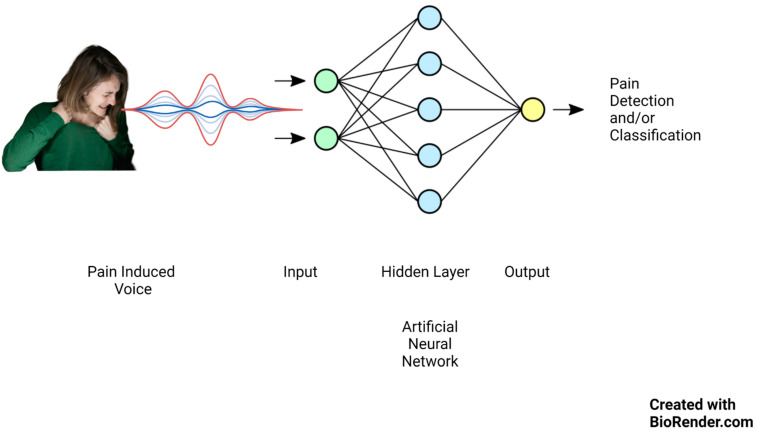
Artificial neural network mechanism of action on pain-induced vocalization.

**Table 1 bioengineering-10-00500-t001:** Reviewed studies’ characteristics.

Author	Subjects	Modality Type	Speech Parameterization Method	AI/ML Type	Model Validation Method	Pain Characteristics	Metric Score
Oshrat et al., 2016 [54]	Patients with spinal cord and/or brain injuries 27 cases (20 male, 7 female) 400 sound files	Audio	OpenSMILE toolkit RASTA-PLP MFCC logMelFreqBand lspFreq	CFS SMO on SVM	Cross-validation and five folds	Not significant (pain levels ≤ 2) Significant (pain levels ≥ 2.5)	CCI ratio 73.75% OpenSMILE 77.25% OpenSMILE+ new features Kappa 39.81% First Group 46.97% Second Group
Tsai et al., 2016 [57]	On-boarding emergency patients 117 cases (205 sound files)	Audio–video Physiological (HR, SBP, DBP) vital sign data Other physiologically relevant results (analgesic prescription and patient disposition)	LLD	2-Class SVM 3-Class SVM Linear regression model (supervised)	Leave one patient out Cross-validation	Binary and tertiary pain severity classification	Accuracy 72.3% Binary Classification 51.6% Ternary Classification
Tsai et al., 2017 [14]	On-boarding emergency patients 63 cases (126 sound files)	Audio–video Physiological (HR, SBP, DBP) vital sign data Other physiologically relevant results	LLD	LSTMs with stacked bottlenecks	Leave one patient out Cross-validation	Binary and tertiary pain severity classification	Accuracy 72.3% Binary Classification 54.2% Tertiary Classification
Li et al., 2018 [7]	On-boarding emergency patients 141 cases 335 sound files (201 male, 134 female)	Audio–video	LLD (MFCC)	MMD-CVAE (unsupervised) linear-kernel SVM	Leave one speaker out Cross-validation	Binary and tertiary pain severity classification	UAR 70.7% Binary Classification 47.4% Tertiary Classification
Tsai et al., 2019 [2]	On-boarding emergency patients 184 cases (323 sound files)	Audio–video Physiological (HR, SBP, DBP) vital sign data other physiologically relevant results	OpenSMILE toolkit LLD	EEMN	Leave one patient out Cross-validation	Binary and tertiary pain severity classification	UAR 70.1% Multimodal Binary 52.1% Multimodal Tertiary
Mohan et al., 2022 [50]	Healthy adults 60 cases (360 sound files)	Audio	LLD (MFCC)	CNN	N/A	Pain Detection	TP 89% to 100% 97.91% Overall Accuracy

Abbreviations: OpenSMILE (Open Speech and Music Interpretation by Large Space Extraction), RASTA-PLP (RelAtive SpecTral PLP), MFCC (mel-frequency cepstral coefficients), logMelFreqBand (logarithmic power of mel-frequency band), LspFreq (Line Spectral Pair Frequencies), LLD (low-level descriptors), CFS (Correlation-Based Feature Selection), SMO (Sequential Minimal Optimization), SVM (Support Vector Machine), LSTM (Long Short-Term Memory), MMD-CVAE (maximum mean discrepancy variational autoencoder), EEMN (emotion-enriched multitask network), CNN (Convolutional Neural Network), CCI (Correctly Classified Instances), UAR (Unweighted Average Recall), TP (True Positive).

## Data Availability

Publicly available datasets were analyzed in this study. This data can be found in the reference section.

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
