# Peer review of "A Review of Voice-Based Pain Detection in Adults Using Artificial Intelligence"

_bioengineering, 2023, doi:10.3390/bioengineering10040500_

Round 1

Reviewer 1 Report

This paper presents AI-based voice analysis can be an effective tool for pain detection in adult patients with various types of pain, including chronic and acute pain. In Table 1, it is the main conclusion. However, That should  be required more comparison results and applications.

Author Response

Point 1: This paper presents AI-based voice analysis can be an effective tool for pain detection in adult patients with various types of pain, including chronic and acute pain. In Table 1, it is the main conclusion. However, that should be required more comparison results and applications.

Response 1: Thank you for your feedback. On Page 7, line 13 (Review of the studies/AI and pain triage), the title has been changed, and more comparison regarding acute and chronic pain is added.

Reviewer 2 Report

The manuscript reported pain by AI. However, the authors did not clear about the content. The only one figure in the manuscript is not nolverty. I do not feel this review paper give clear idea to readers. I recomment to reject.

Author Response

Point 1: The manuscript reported pain by AI. However, the authors did not clear about the content. The only one figure in the manuscript is not nolverty. I do not feel this review paper give clear idea to readers. I recomment to reject.

Response 1: Thank you for your feedback.

To provide a more comprehensive review and stronger study, in the text, we tried to include more information by comparing the metric scores (page 12, line 30) and acute and chronic pain types (page 7, line 13). In addition, we added more information about preprocessing stages of audio data (pages 7, line 34), (page 8, line 13), (page 9, lines 18 and 37), and (page 10, line 23).

We added the validation methods of the models with more description of cases involved in the studies in Table 1.

In addition, we submitted the graphical abstract to the journal and edited the English proficiency of the manuscript extensively.

Reviewer 3 Report

Voice-Based Pain Detection is a good problem addressed by the authors. I found it useful and willing to recommend. Some changes are as follows:

(1) In Section 2, there is only one Subsection so number of sebsection 2.1 can be avoided

(2) As this is review paper so authors may increase the number of references

(3) Accuracy and reliability of voice-based pain detection should be addressed in the work

(4) The ethical implications of using voice-based pain detection methods is required particularly in terms of data privacy and consent. 

(5) Try to include the Clinical application of voice-based pain detection  in your work

Author Response

Point 1: In Section 2, there is only one Subsection so number of sebsection 2.1 can be avoided

Response 1: Thank you for your feedback. In Section 2 number of subsection 2.1 was deleted.

Point 2: As this is review paper so authors may increase the number of references

Response 2: Thank you for your advice. We increased the number of references to 83.

Point 3: Accuracy and reliability of voice-based pain detection should be addressed in the work

Response 3: Thank you for your feedback. On Page 12 (summary of findings), line 30 metric scores were added.

Point 4: The ethical implications of using voice-based pain detection methods is required particularly in terms of data privacy and consent. 

Response 4: Thank you for your feedback. While ethical implications are one of the most critical subjects in AI studies, due to its broad nature, it’s out of the scope of this review. We mentioned this on Page 13, line 45 (limitations) of the review.

Point 5: Try to include the Clinical application of voice-based pain detection in your work

Response 5: Thank you for your comment. We have modified the title and replaced ‘Potential implications’ with ’Clinical applications’ on page 12, line 39. Please let us know if any specific matter should be addressed in the text.

Reviewer 4 Report

The authors present a literature review (through the analysis of only five papers) synthesizes the current state of research on the use of voice recognition and voice analysis for pain detection in adults, with a specific focus on the role of artificial intelligence (AI) and machine learning (ML) techniques.

I understand that the research area covered by this paper is a niche area, but a review based on only five papers is hard to find in the literature.

The following authors may consider my observations:

1]     Integrate the values of the accuracy indexes of the models that appear in the trained models present in the papers examined for the study.

2]     Report, if any, the way in which the dataset was divided to train and test the networks in the trained models present in the considered papers.

3] Indicate whether particular treatments have been made on the data (ex. normalized features) before their use in model training. Also indicate how the dataset used to train the network is made up (number of subjects and number of tracks, if possible, also indicate how many subjects are men and women).

4] Correct some punctuation errors in some difficult to understand sentences.

From my point of view, point 2 and 3 are fundamental for the final quality of the model.

Even with a small dataset, results can vary depending on how you prepare and use the data.

Author Response

Point 1: Integrate the values of the accuracy indexes of the models that appear in the trained models present in the papers examined for the study.

Response 1: Thank you for your advice. On Page 12, line 30 (summary of findings), metric scores are added. Please let us know if more or different information should be addressed.

Point 2: Report, if any, the way in which the dataset was divided to train and test the networks in the trained models present in the considered papers.

Response 2: Thank you for your comment. We have modified Table 1 and added the column for Validation Method.

Point 3: Indicate whether particular treatments have been made on the data (ex. normalized features) before their use in model training. Also, indicate how the dataset used to train the network is made up (number of subjects and number of tracks, if possible, also indicate how many subjects are men and women).

Response 3: Thank you for your comment. We have modified the below sections of the manuscript and added the preprocessing steps for all the studies:

Page 7 (Review of the studies/ AI and pain triage) line 34

Page 8 (Review of the studies/ AI and pain triage) line 13

Page 9 (pain as a complex affection), line 18

Page 9 (Can speech prosody act as a biosignal?) line 37

Page 10 (Age, Gender, Pain) line 23

We also added the available subjects and number of tracks as a column to Table 1.

Point 4: Correct some punctuation errors in some difficult to understand sentences.

Response 4: Thank you for your advice. We have modified the text and edited the English proficiency of the manuscript extensively.

Round 2

Reviewer 4 Report

The changes made by the authors improve the quality of the paper and make it worthy of publication.